# Extended Use of Topical Efinaconazole Remains Safe and Can Provide Continuing Benefits for Dermatophyte Toenail Onychomycosis

**DOI:** 10.3390/jof10090620

**Published:** 2024-08-30

**Authors:** Aditya K. Gupta, Elizabeth A. Cooper

**Affiliations:** 1Division of Dermatology, Temerty Faculty of Medicine, University of Toronto, Toronto, ON M5S 1A8, Canada; 2Mediprobe Research Inc., 645 Windermere Rd., London, ON N5X 2P1, Canada; lcooper@mediproberesearch.com

**Keywords:** onychomycosis, tinea unguium, dermatophyte, antifungal agents, efinaconazole

## Abstract

Introduction: Efinaconazole 10% topical solution labeling for onychomycosis describes phase III trials of 12 months of treatment; the slow growth of onychomycotic nails suggests a longer treatment period may increase efficacy. We present here the first evaluation of extended use of efinaconazole 10% topical solution for up to 24 months. Materials and Methods: Enrolled patients (n = 101) had one target great toenail with mild to moderate distal lateral subungual onychomycosis and applied efinaconazole 10% topical solution to all affected toenails once daily for 18 months (EFN18) or 24 months (EFN24). Efficacy and safety were evaluated at each visit by visual review and mycology sampling. Results: Regarding the target toenail for patients treated for 24 months (EFN24), mycological cure (negative microscopy and culture) was 66.0% at Month 12, increasing to 71.7% at Month 24; effective cure (mycological cure and ≤10% affected nail) was 13.2% at Month 12, rising to 22.6% at Month 24. Mild to moderate application site reactions (symptoms of erythema/scaling) were the only efinaconazole-related reactions, in eight patients (7.9%). No systemic efinaconazole events or drug interactions were found. Patients aged 70 years or more had similar efficacy to younger patients at all time periods and did not show any increased treatment risks. Thinner nails exhibited better clearance versus thicker nails. A higher proportion of patients with *Trichophyton mentagrophytes* complex infection experienced application site reactions (35.7%), and a higher effective cure was found at Month 24 versus *T. rubrum* patients. Conclusion: There is a trend of increasing mycological cure and effective cure beyond Month 12 to Month 24, without an increased safety risk. The enrolled population in this trial was significantly older than in the phase III trials, with a greater degree of onychomycosis severity; however, increased age did not appear to reduce the chance of efficacy to Month 24 in this study. Our data suggest that lack of ability to clear nail dystrophy remains a significant problem for patients, rather than any lack of efinaconazole action over long-term treatment periods.

## 1. Introduction

Onychomycosis is a chronic fungal nail infection which, though it appears initially as merely a cosmetic issue, can progress to significant thickness and deformation leading to difficulty in trimming, fitting shoes, and walking, while also being a possible source of concomitant ulceration and foot infection [1,2,3,4,5,6]. Though oral antifungals such as terbinafine and itraconazole have shown good success in treating onychomycosis, these agents are associated with risks of drug interaction and systemic adverse effects such as hepatotoxicity, both of which make oral agents undesirable for a majority of older onychomycosis patients who most need effective therapy to maintain health and mobility. Topical antifungals may be the only safe alternative for these patients. 

Efinaconazole 10% topical solution (Jublia^®^) was approved in 2013 in Canada and in 2014 in the USA for mild to moderate dermatophyte toenail onychomycosis [7,8]. Efinaconazole 10% solution accumulates to high concentrations in the nail plate and nail bed after continuous application for 14 days, well above the MIC of dermatophytes that cause onychomycosis, in addition to showing a broad spectrum of action against dermatophytes, yeasts, and non-dermatophyte molds (NDMs) [9,10]. The phase III studies of efinaconazole 10% solution reported on a 48-week once-daily treatment period. Despite a promising level of mycological cure, more than half of the treated patients were left with visual signs of infection in the target toenail after the 48-week treatment [11,12]. The low clinical cure may reflect slowed outgrowth of nail dystrophy rather than lack of mycological efficacy; it was hypothesized that allowing the toenail an extended period for outgrowth while continuing use of efinaconazole may improve clinical cure rates. To investigate this possibility, we performed the first study of safety and efficacy of efinaconazole 10% solution once daily for up to 24 months in cases of mild to moderate toenail distal lateral subungual onychomycosis (DLSO). 

## 2. Materials and Methods

A phase IV single-site investigator-initiated Canadian trial enrolled adult patients showing mild to moderate DLSO (20–50% of the toenail affected) in a target great toenail, with at least 1mm clear nail at the proximal nail fold and thickness of ≤3 mm. Diagnosis was confirmed by dermatophyte growth in culture, identified morphologically by an experienced mycologist. Additionally, patients agreed to abstain from any other nail/foot treatment for the duration of this study except the study product as directed for use. This study was approved by an Ethics Review Board (Advarra IRB; formerly IRB Services) and written informed consent was obtained from all participating subjects. Health Canada approval was not required as the once-daily usage fell within the Notice of Compliance for efinaconazole 10% solution.

Eligible study subjects were supplied with the commercial formulation of efinaconazole 10% *w*/*w* in a clear, low surface tension vehicle solution for topical application. The efinaconazole vehicle formulation contains the following inactive ingredients: alcohol, butylated hydroxytoluene, C12- 15 alkyl lactate, citric acid, cyclomethicone, di-isopropyl adipate, disodium edetate, and purified water.

The first 6 months of study was a double-blinded vehicle-controlled once-daily application phase, followed by 18 months of open-label efinaconazole 10% solution use (24 months total study participation). During the initial phase of research (January 2016 to April 2018), eligible subjects were randomized 1:1 into EFN24 group (once-daily efinaconazole 10% solution for 24 months: the first 6 months were given in a double-blind fashion, then given as open-label after Month 6) or EFN18 group (once-daily efinaconazole vehicle for first 6 months given in double-blind fashion followed by open-label once-daily efinaconazole 10% solution for 18 months). Vehicle/blinded drug supply issues mandated a change in treatment plan as of May 2018, and remaining patients enrolled in this study were provided with open-label topical efinaconazole 10% solution from Day 1 to 24 months (EFN24). Efinaconazole 10% solution/vehicle solution was to be applied topically once daily on all infected toenails including the ‘target’ great toenail, with application over the entire nail surface, cuticle, and along/under the distal edge of the toenail. 

Enrolled subjects returned for visits at Months 3, 6, 9, 12, 16, 20, and 24. Subject safety was reviewed at all visits. At each visit, the target great toenail was assessed for onychomycosis area, length, and thickness measures by a single site assessor. Mycology sampling of the target toenail was performed at each efficacy visit timepoint by a single staff member experienced in nail collection. Nail sample portions were prepared with potassium hydroxide (KOH) and Remel^™^ BactiDrop Calcofluor White, then examined under fluorescence using the Zeiss Axiovert 200 inverted epifluorescence emission microscope. Remaining nail sample portions were plated on Sabouraud dextrose agar plates with chloramphenicol/gentamicin to inhibit bacterial growth (CG SDA; Bio-Media Unlimited Ltd., Woodbridge, ON, Canada) and chloramphenicol/cycloheximide/gentamicin to inhibit growth of bacteria and non-dermatophyte fungi (CCG SDA; Bio-Media Unlimited Ltd, Woodbridge, ON, Canada.), and thereby increase the likelihood of dermatophyte growth/detection. 

Safety evaluation reviewed all adverse events reported to the site during the trial period. The frequency of any events possibly or probably related to efinaconazole were tabulated and reviewed to determine if the events suggest there are long-term safety issues developing with long-term efinaconazole use.

Efficacy to Month 24 was evaluated by calculating group proportions of the following outcomes: mycological cure (MC; negative fluorescent microscopic examination and negative culture), and effective cure (EC; mycological cure with ≤10% visual clinical involvement of the target toenail). Statistical review used the PSPP software program (GNU pspp 2.0.0-g5b54d1) to examine distribution of efficacy proportions between groups by Chi-square testing with a 95% confidence interval. Statistical review of difference between mean values used *t*-test or ANOVA (as applicable to the number of categories compared), with 95% confidence interval.

## 3. Results

### 3.1. Subject Demographics

Fifty-five (55) patients were randomized 1:1 into blinded treatment groups as of April 2018 (EFN24: n = 25; EFN18: n = 30). From May 2018 onward, a further 46 patients were provided with open-label topical efinaconazole 10% solution every day for 24 months (EFN24), for a total of 101 patients enrolled. 

Of the 101 patients enrolled, 69 patients completed 24 months of treatment per the protocol (EFN24: n = 53; EFN18: n = 16). The demographics of the completed patients are shown in Table 1. The average baseline affected area is similar between the two groups, and most patients are aged 60 years or older. *Trichophyton rubrum* was the most frequently reported dermatophyte organism, as expected. Most patients showed multiple toenail involvement, but few patients had concomitant fingernail infection. 

### 3.2. Safety and Adverse Events

In total, 59 adverse events (AEs) were reported in 46 of the 101 enrolled patients. Of these events, 40 were non-serious and considered unrelated to study treatment. Events in this category were mild to moderate, and within the scope of events expected for an older patient population. For serious adverse events (SAEs), nine events were reported, occurring in eight patients (Table 2): the events were within the ‘normal’ spectrum for the study population and none of these events were considered related to efinaconazole 10% solution application. Most patients were able to complete this study to Month 24 despite these AEs.

There were 10 events considered as reactions possibly, probably, or likely related to use of the efinaconazole treatment, experienced by eight total patients (Table 2). Application site reactions were mild to moderate only, with affected patients reporting symptoms of erythema and scaling typical of ‘dermatitis’, as reported in the pivotal efinaconazole phase III trials [11]. No systemic reactions occurred in association with the efinaconazole 10% solution application. No patients reported reactions during the vehicle-use period. A majority of the reported events occurred within the first 9 months of efinaconazole 10% solution application, i.e., in the ‘labelled’ period of use. For reactions that developed after Month 12, two patients reported application site trauma not related to study participation which may have predisposed them to efinaconazole 10% solution reaction (Table 2—2 patients). Prolonged once daily use of efinaconazole 10% solution from M12 to M24 does not appear to increase the risk of an application site reaction.

Of the eight patients with possible efinaconazole reaction, five of eight (62.5%) had infection organisms identified as *T. mentagrophytes* complex, though only 13.9% of the enrolled patients were found to have causative dermatophyte organisms within the *T. mentagrophytes* complex. This high proportion is curious, and we were unfortunately not able to investigate further speciation within the *T. mentagrophytes* complex. It may be valuable to pay more detailed attention to the relation of reaction/cure rates in association with dermatophyte species in future research.

### 3.3. Efficacy Outcomes

Efficacy criteria were evaluated at Months 6, 12, 20, and 24. Efficacy at Month 6 represents only vehicle use in the EFN18 group (n = 16), with months 12, 20, and 24 follow-up in EFN18 group representing ongoing efinaconazole 10% once daily use for up to 18 months. The main efficacy results are presented in Figure 1.

A moderately high rate of mycological cure was found after 6 months of active efinaconazole use, and ongoing to Month 24 (Figure 1). After 12 months of efinaconazole 10% solution, the MC was 60%, similar to the phase III trial efficacy (53–55%) [11,12]. The MC of vehicle for 6 months (EFN18 at Month 6) was 20%, similar to the vehicle MC of the phase III trials at week 52 (17%) [11,12]. MC in the EFN18 group was able to achieve comparable MC to EFN24 after 6 months of active efinaconazole use to M12, and ongoing to M24 follow-up. There appears to be some increase in MC in both groups at M24 compared to M12.

Effective cure increases from M12 to M24 in both groups, indicating continued improvement in target nails beyond the 12-month period with continued efinaconazole 10% once daily use (Figure 1). The EFN18 group was able to achieve similar outcome measures to the EFN24 group at M24. Figure 2 shows successful clinical outcomes of a selection of patients using efinaconazole for 24 months.

Though a single great toenail was chosen as a target for logistical ease in the trial, a majority of our patients had many other toenails showing signs of dystrophy (mean = 5.1 toenails, Table 1), and these toenails were assessed visually during the trial along with the target toenail. The M24 assessment showed that more than 80% of 69 patients had at least one toenail cleared (Table 1). Overall, 133/333 toenails (39.9%) showed resolution of dystrophy at M24. Though onychomycosis cure cannot be definitively assumed for these non-target toenails, it appears that dystrophy resolution in the non-great toenails may be higher than that of the great toenails after 24 months of efinaconazole.

The enrolled population in this trial is significantly older than that in the phase III trials which restricted age to 70 years or less [11,12]. Considering efficacy by patient age in the EFN24 group only, the EC rates show some increase from M12 to M24 for the ≥70-year subgroup, as well as the younger subset, and reached similar levels at M24 regardless of age (Figure 3). Increased age does not appear to be a negative factor for obtaining improvements with efinaconazole.

Increased thickness of nails presents a significant barrier to the penetration of topical drugs and their ability to act on subungual fungi. Considering nail thickness in the EFN24 group only, EC at M24 was significantly higher in nails ≤ 1mm (*p* < 0.05) versus thicker nails (Figure 4); thicker nails did achieve some benefit from prolonged efinaconazole beyond Month 12 but not to the same degree as thinner toenails. The addition of routine nail care such as nail thinning may be needed to increase efficacy.

When outcomes were considered by infecting organism in EFN24 patients, we were surprised to again note apparent differences between the *T. mentagrophytes* complex and *T. rubrum* groups. *T. mentagrophytes* complex infection showed higher rates of mycological and effective cure versus *T. rubrum* (Figure 5); the *T. mentagrophytes* complex population is small, and statistical evaluation did not always show a significant difference, but the 100% MC suggests that efinaconazole may be a highly effective option for the treatment of this dermatophyte group. Further speciation within the *T. mentagrophytes* complex group would be useful in future to determine if there are species-specific actions of efinaconazole.

## 4. Discussion

The data presented here represent the first assessment of a 24-month efinaconazole 10% solution use period and demonstrate the safety and efficacy of such a regimen. At Month 24, the MC and EC in all patients applying efinaconazole 10% solution for the full study period (EFN24) was 71.7% and 22.6%, respectively. Patients switching to the active drug at Month 6 (EFN18) also showed good MC and EC at Month 24. Increases in MC and EC from Month 12 are noted at Month 24, indicating some benefit to prolonged treatment to Month 24, with no increased treatment risks.

In phase III trials, the mycological (MC) and complete cure rates (CC) of efinaconazole 10% solution were 53–55% and 15–18%, respectively [8,11,12]. Our Month 12 outcomes for MC are higher than these efinaconazole pivotal trials, but it appears our population has not had as much success as the controlled phase III populations in clearing visible signs of infection at Month 12. As this is a phase IV study, the patients eligible for enrollment are more representative of the ‘real-world’ onychomycosis population, and our enrolled population was somewhat more severely affected than the phase III studies. Our population included onychomycosis which penetrated more proximally into the nail plate of the target toenail (lowest proximal extent) than in the phase III populations, with almost 50% of patients having less than 3 mm clear nail at enrollment. It is expected that such severity would require an overall longer period of outgrowth/lower cure rate at similar time points relative to less severe populations. We also had a significant number of patients greater than 70 years of age. Increased age is a further burden for nail clearance, as there is a decrease in immune competence, peripheral circulation, and slower outgrowth of toenail [1,13,14]. The MC and EC at Month 24 did not demonstrate any significant difference between younger and older groups, suggesting that the prolonged use of efinaconazole may be useful for more severe infections in patients of any age.

The increase in efficacy from Month 12 to Month 24 is attributed to an increased number of patients being able to grow out the infected target nail by maintaining fungicidal or fungistatic action with the extended use of efinaconazole beyond Month 12. Though the increase in EC from Month 12 to Month 24 is not as high as may be desirable, there are indications this lack of improvement may be due to ongoing nail dystrophy and lack of ability to clear the dystrophy, rather than any lack of efinaconazole action. It was noted by the site that patients failing to achieve MC at Month 24 failed by microscopy only; no patients showed positive dermatophyte culture at Month 24, regardless of outcome status. Of the 336 post-enrolled cultures, only 8 samples (8/336, 2.4%) grew any type of fungi after starting efinaconazole (1-dermatophyte; 3-Candida spp.; 4-contaminant NDMs). Three of these positive cultures were in two subjects terminating efinaconazole use early due to AEs, but continuing in safety follow-up to Month 24. Efinaconazole appears to effectively block the growth of dermatophytes and other fungi in culture from an early stage and throughout use of efinaconazole to Month 24, and regardless of nail thickness, likely due to drug retention in the sampled nail. Though it has been reported that a toenail may need more than 12 months to grow out, our data demonstrated very few nails outgrowing in the first 12 months, with only a small addition to Month 24. The ability of a toenail to clear itself of infection within 12–24 months after significant fungal infection appears to be relatively low in most cases, and some nail dystrophy may never resolve. Additional nail care methods to increase nail outgrowth may be needed to see optimal efinaconazole efficacy. While dystrophy remains, the risk of relapse or reinfection is high.

It is a limitation of clinical trials that typically only target toenail efficacy is reviewed; however, for a patient using antifungal treatment, it is the outcome of all affected toenails being treated that is of concern. We have attempted to characterize efficacy for all affected toenails in this trial by providing visual assessments of any affected toenails at follow-up visits. As non-target toenails will mostly be non-great toenails, with much smaller toenail areas, it may be expected that cure could occur at a higher rate, and more rapidly, in these non-great nails compared to great toenails. We have noted that a high proportion of dystrophic toenails were clinically cured of infection to Month 24, but it would also be relevant to review efinaconazole mycological efficacy more thoroughly in future as an option for treatment of non-great toenails.

The adverse effects with 18–24 months of efinaconazole 10% solution use were similar in nature to the adverse events reported in the phase III trials, and a majority of events occurred in patients <70 years old. The older subset of patients is the population most in need of non-oral antifungal treatment options, and being able to confirm safety for these patients is critical. In this study, there were no systemic adverse events associated with efinaconazole 10% solution, no drug interactions noted, and all treatment-related adverse events were localized to the site of action. This long-term study provides evidence that efinaconazole 10% solution may be used safely over 18–24 months, including elderly patients. In general, our safety data do not show any increased safety risk for an elderly subset.

The causative organism in toenail onychomycosis in North America is generally *Trichophyton rubrum* and to a lesser extent *T. mentagrophytes* complex. Our study population reflected this general demographic finding. The effective cure was significantly higher in the *T. mentagrophytes* complex patients versus *T. rubrum* at Month 24 (patients using efinaconazole for 24 months only). Of the 14 patients enrolled with primary *T. mentagrophytes* complex infection, 5 patients (5/14, 35.7%) reported some degree of efinaconazole reaction. There are many other patient factors that may be relevant to these findings and these were not investigated here, but our data suggest a more active response by the patients on several fronts when *T. mentagrophytes* complex is the causative organism group, and more investigation into species-specific outcomes is warranted for efinaconazole.

In summary, this trial demonstrates the effectiveness and safety of efinaconazole 10% solution for once-daily use beyond 12 months to 24 months; application appears to remain safe even for elderly patients. Analysis of the interim data demonstrate a trend of increasing MC and EC beyond 12 months’ use, without any increased risk of adverse effects. Failure to achieve cure may be attributable to lack of nail outgrowth more so than lack of efinaconazole penetration/action, and suggests that patients may benefit from longer use of efinaconazole in conjunction with more nail care practices.

## Figures and Tables

**Figure 1 jof-10-00620-f001:**
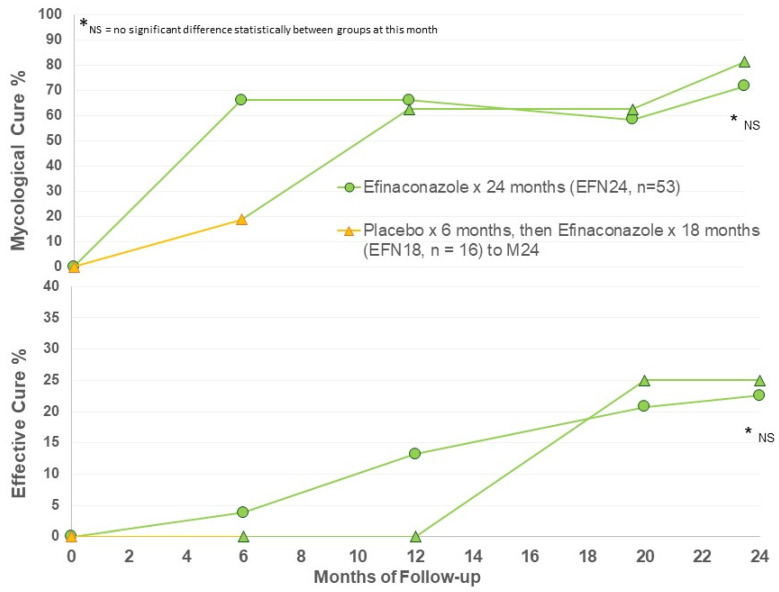
Mycological and effective cure rates by treatment group.

**Figure 2 jof-10-00620-f002:**
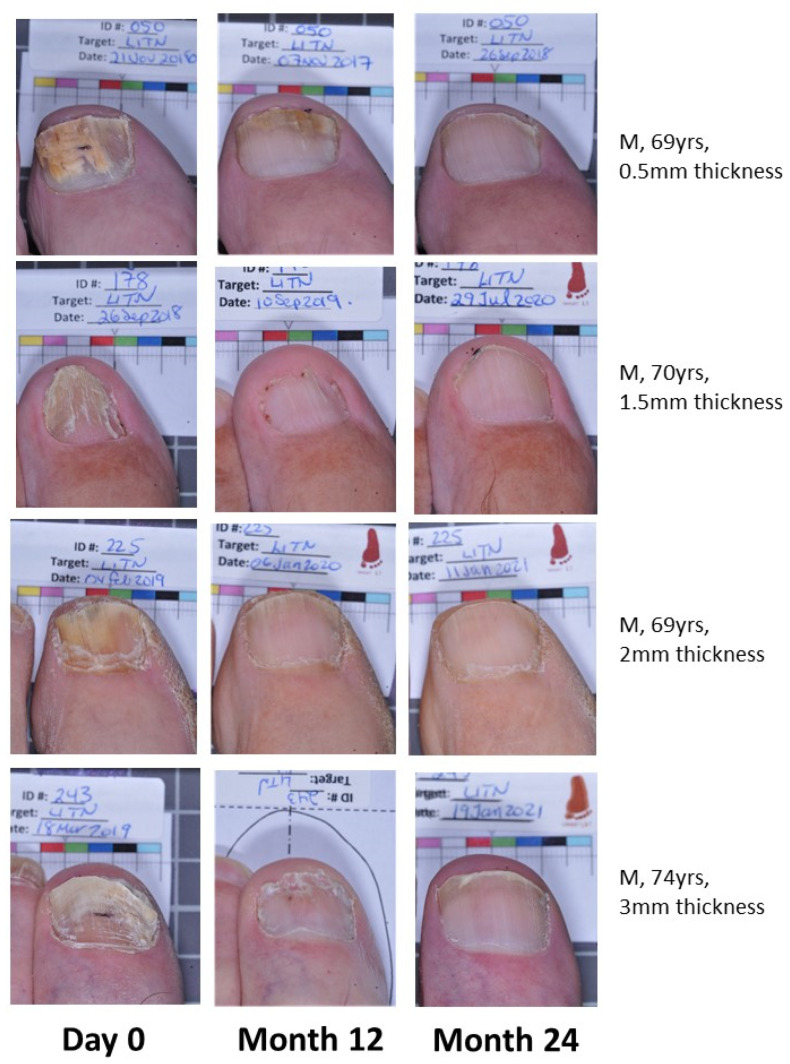
Patient photos (EFN24) demonstrating onychomycosis cure at Month 24.

**Figure 3 jof-10-00620-f003:**
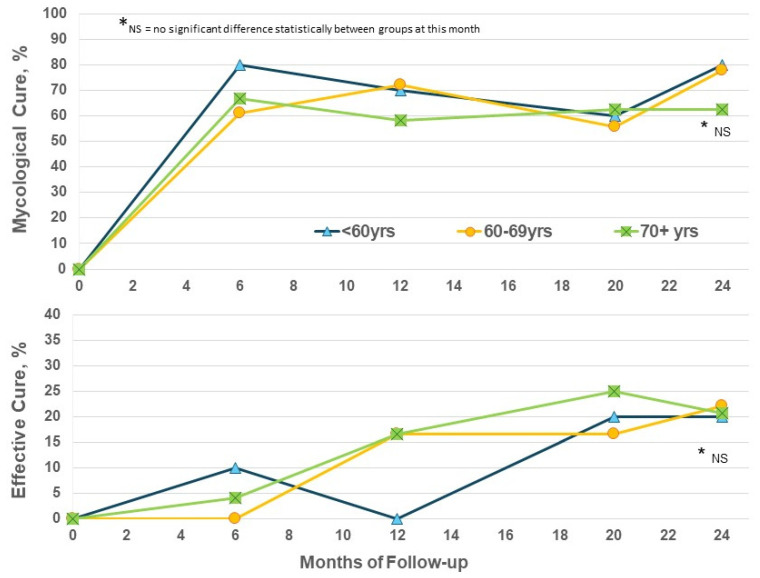
Efinaconazole for 24 months (EFN24 ONLY): cure rates by age.

**Figure 4 jof-10-00620-f004:**
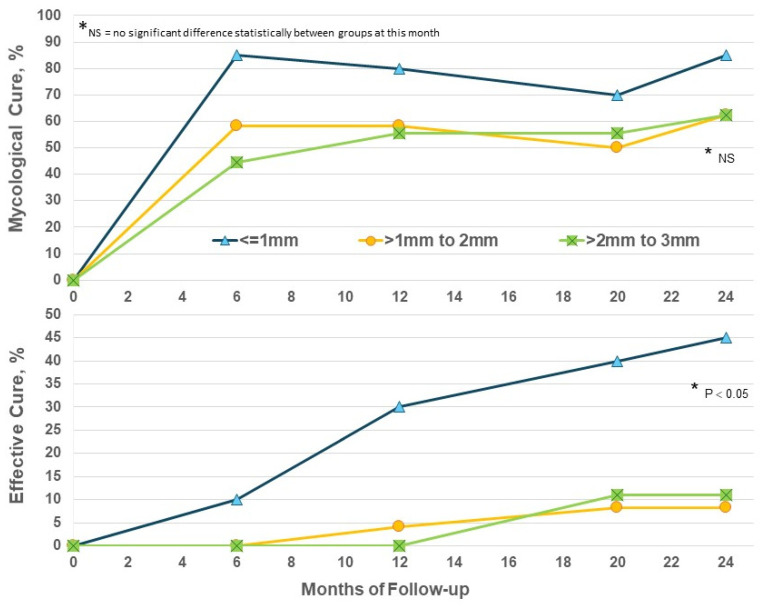
Efinaconazole for 24 months (EFN24 ONLY): cure by nail thickness.

**Figure 5 jof-10-00620-f005:**
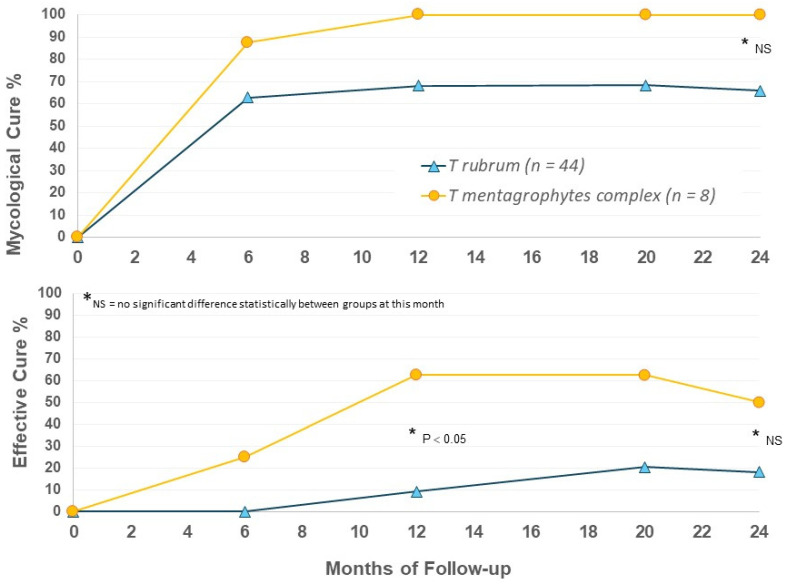
Efinaconazole for 24 months (EFN24 ONLY): cure rates by organism.

**Table 1 jof-10-00620-t001:** Demographics summary—patients completing daily efinaconazole to Month 24.

	Efinaconazole 10% Solution 18-Month Use (EFN18)	Efinaconazole 10% Solution 24-Month Use (EFN24)	Total Number of Patients
‘N’ patients enrolled-Day 0	30	71	101
Lost to follow-up:	5	8	13
Withdrew consent:	0	6	6
Moved:	1	0	1
Efinaconazole 10% solution reaction:	2	0	2
SAE:	1—lung cancer	0	1
AE:	1—ingrown TN	0	1
Deviations-data not valid for analysis	4	4	8
Total withdrawn from M24 analysis:	14	18	32
‘N’ patients completing to M24; assessed for efficacy	16	53	69
For N = 69 patients completing Month 24:			
Sex ^NS^	Female = 1 (6.3%); Male = 15 (93.8%)	Female = 9 (17%); Male = 44 (83%)	Female = 10 (14.5%); Male = 59 (85.5%)
Diabetes ^NS^	3/16 (18.7%)	11/53 (20.8%)	14/69 (20.3%)
Average age, yrs (Min, Max)	61.43 yrs ± 9.66 (45, 77)	67.55 yrs ± 10.81 (24, 89)	66.13 yrs ± 10.80 (24, 89)
Distribution: ^NS^ <45 yrs 40–59 yrs 60–69 yrs 70+ yrs	<45 yrs: 0 pt 40–59 yrs: 7 pts 60–69 yrs: 6 pts 70+ yrs: 3 pts	<45 yrs: 1 pt 40–59 yrs: 10 pts 60–69 yrs: 18 pts 70+ yrs: 24 pts	<45 yrs: 1 pt 40–59 yrs: 17 pts 60–69 yrs: 24 pts 70+ yrs: 27 pts
Average # of dystrophic TN per subject (Total N of dystrophic TN)	5.88 ± 3.14 (94)	4.51 ± 2.89 (239)	5.1 (333)
# of patients with FN involvement	1/16 (6.3%)	1/53 (1.9%)	2/69 (2.9%)
Average area involved, % ^NS^	42.81% ± 10.32	42.36% ± 10.99	42.46% ± 10.77
Dermatophyte detected: ^NS^	*T. rubrum*: 14 (87.5%) *T. mentagrophytes* complex: 1 (6.3%) *Trichophyton* sp.: 1 (6.3%)	*T. rubrum*: 44 (83.0%) *T. mentagrophytes* complex: 8 (15.1%) *T. tonsurans*: 1 (1.9%)	*T. rubrum*: 58 (84.1%) *T. mentagrophytes* complex: 9 (13.0%) *T. tonsurans*: 1 (1.4%) *Trichophyton* sp.: 1 (1.4%)
At least 1 TN cleared by M24 of treatment (target/non-target)	14/16 (87.5%)	43/53 (81.1%)	57/69 (82.6%)
Total N of TN cleared (% of affected TN cured)	42/94 (44.7%)	91/239 (38.1%)	133/333 (39.9%)

EFN = efinaconazole; TN = toenail; N = number; FN = fingernail; *T.* = Trichophyton; SAE = serious adverse event; AE = adverse event; M24 = Month 24 of continued, once-daily study treatment use; ^NS^ = no statistically significant difference between treatment groups.

**Table 2 jof-10-00620-t002:** Safety events with efinaconazole 10% daily use to Month 24.

Serious Adverse Events (N = 9, in 8 patients)					
Completed study:	Myocardial infarction: 3 pt - Male, 53 yrs; EFN18 - Male, 76 yrs; EFN24 - Male, 73 yrs; EFN24		Bilateral pulmonary emboli: 1 pt - Male, 65 yrs; EFN18	Possible bradyarrhythmia: 1 pt - Male, 77 yrs; EFN18	Surgical repair of umbilical hernia: 1 pt - Male, 57 yrs; EFN24
Early termination:	Lung cancer—terminal stage: 1 pt (Male, 75 yrs; EFN18)				
Lost to follow-up:	Blood clot in heart; accidental lorazepam overdose (2 events; 1 subject—Female, 60 yrs; EFN18)				
2. Possible efinaconazole 10% solution reactions—8 patients with 10 ‘possible/probable/likely’ application site reactions reported Mild to moderate grading of all reactions; similar symptoms found to varying degree in all affected patients: erythema, scaling (‘dermatitis’)					
Reaction starting between Month 0 to Month 3 of efinaconazole active treatment: 2 pts		Temporary interruption of efinaconazole 10% solution for healing; efinaconazole 10% solution restart with a return of symptoms and signs; intermittent use adopted to control symptoms, and allowed application to continue. Male, 69 yrs-*T rubrum*-EFN24; start Month 0-LTF after Month 12 visit Male, 55 yrs-*T rubrum*-EFN24; start Month 3—completed to Month 24; intermittent mild reaction throughout study, resolved with temporary treatment interruption; MC at Month 24, but no reduction in affected onychomycosis area ○ Same pt also developed a similar FN application site reaction at Month 9; resolved upon stopping efinaconazole 10% solution. Efinaconazole 10% solution permanently discontinued from application to pt FNs.			
Reactions starting after switch from vehicle to active treatment: 2 pts		Efinaconazole 10% solution permanently withdrawn: Male, 43 yrs-*T. mentagrophytes* complex -EFN18; start Month 9-treatment permanently withdrawn; ET Male, 55 yrs-*T rubrum*-EFN18; start Month 12-treatment permanently withdrawn; ET			
Reaction starting Month 9 of efinaconazole treatment: 1 pt		Efinaconazole 10% solution permanently withdrawn: Male, 84 yrs-*T. mentagrophytes* complex-EFN24; resolution of symptoms after stopping treatment; patient opted for no further treatment, remained in study to Month 24 without further study treatment – no further reaction; no EC (long-term safety FU)			
Mild toe web reaction: 1 pt		Attributed to poor application technique; efinaconazole 10% solution continued with more attention to application. Male, 48 yrs-*T. mentagrophytes* complex-EFN24; start Month 6-resolved with more attention to application; completed to Month 24 ○ Same pt developed application site reaction at Month 16 period; temporary interruption for healing; able to resume efinaconazole 10% solution daily use; completed to Month 24—no EC			
Reactions after application site trauma events: 2 pts		Symptoms resolved with efinaconazole 10% solution interruption: Insect bite at Month 22: Female, 74 yrs-*T. mentagrophytes* complex-EFN24: EC prior to event, bite/reaction resolved with treatment interruption; patient opted to stop efinaconazole 10% solution permanently; completed FU to Month 24, remained EC Hiking boot bruising/blistering at Month 16; Male, 57 yrs-*T. mentagrophytes* complex-EFN24: trauma was exacerbated by efinaconazole use; reaction resolved upon treatment interruption, but returned upon restarting application—efinaconazole 10% solution permanently withdrawn at Month 20. Followed to Month 24 without further study treatment—no further reaction, no EC (long-term safety FU)			
Reaction reports: 8/101 (7.9%)		yrs = years old; EFN = efinaconazole; ET = early termination; FU = follow up; FN = fingernail; LTF = lost to follow-up; EC = Effective Cure			

## Data Availability

The data presented in this study are available on request from the corresponding author due to ethics and patient privacy concerns.

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
