# Peer review of "Extended Use of Topical Efinaconazole Remains Safe and Can Provide Continuing Benefits for Dermatophyte Toenail Onychomycosis"

_jof, 2024, doi:10.3390/jof10090620_

Round 1

Reviewer 1 Report

Gupta and Cooper present a timely manuscript on the clinical efficacy of efinaconazole 10% topical solution for the treatment of onychomycosis. The work seems very readable and contains relevant results. The strength of the phase IV study that the number of the patients were higher than in the phase III trials with 12-month treatment, and patients were older. Moreover they used longer treatment period without relevant side effects.

I only have some questions for the authors

1. How identified the fungi grew on special medium (microscopic morphology or molecular method)?

2. Regarding that frequently many nails are affected in onychomycosis, what is the cost of the 2 years treatment?

3. Is there any information on the efficacy of efinaconazole 10% topical solution with thickness of > 3mm?

Reviewer 2 Report

The presented article describe a study of use of topical efinaconazole for treatment of toenail onychomycosis of around 70 patients. The authors suggests an extended use of efinaconazole treatment.

Major points:

The authors do not clarify sufficiently the difference between mycological cure and effective cure. Mycological cure means that culture and microscopy were negative. Most effects of efinaconazole on fungal curation are seen in the first six-month treatment period with efinaconazole. Effective cure means nail dystrophy disappears but it is not clear if it is always a sign for infection or a cosmetic problem. If nail infection is healed before nail dystrophy disappeared, a prolonged efinaconazole treatment is not useful. Long-term treatment is also a risk factor for establishing azole resistant fungal cultures. Isolates of the T. mentagrophytes complex as well as T. rubrum have developed several strategies to overcome azole treatment like efflux mechanism due to increased expression of transporters or overexpression of the target protein lanosterol-14 alpha demethylase.

If nail dystrophy is an infection sign, why the authors were unable to isolate fungal cultures at later time-points and microscopy failed also. Fungal cultivation failed sometimes but methods like Real-Time PCR can be done easily with nail material and show increased sensitivity.

Minor points:

Author names

In my opinion, it is not necessary to describe the scientific degrees

Abstract: Line 22-30. The authors should avoid abbreviation in the abstract. The sentences are difficult to understand.

Materials and methods

Line 73: efinaconazole 10% solution. The authors should give a precise description. Do the authors mean 10g efinaconazole per 100 ml of an unknown solvent? Azole are lipophilic compounds, 100mg/ ml will not solved in water like solutions. The authors should specify the solvent used also as vehicle control.

Line 87 “visits at months 3, 6, 9, 12, 16, 20, 24”

The presented times differ from periods presented in line 146 (6, 12, 18, 24 months).

Line 91 “(KOH) fluorescence microscopy”.

KOH show no fluorescence; I miss a citation of the method, the chosen fluorochrome and microscope specifications (emission, excitation, etc.).

Line 92 “CG/CCG agar” The authors should specify the manufacturer and the selection principle.

There is no hint how species identification was made, do the author use an RT-PCR detection procedure?

Species determination only from microscopy and culture morphology has limitations and especially the different species of the T. mentagrophytes complex can be identified only with molecular methods.

Line 103: “statistical review” the authors should describe in more detail which program were used.

Results: Table1  I don’t understand why the category “sex” were named as not statistically significant. The ratio of females and males do not represent a 50/50 distribution and especially the 18 months group has a ratio of 1/15. The authors should explain their statistical analyses.

The authors should avoid equal abbreviations for different words (M = month and male)

Table 2

The letter size differ from the other text and it is difficult to read this table. The authors should present the table in the correct form and avoid equal abbreviations.

Line 127 page numbering  starts at 1 again

Figure 1, 3 and 4

As before discrepancy between periods presented in line 146. Measuring points were located at 20 months in all figures.

Line 151: “Good mycological success” Who is successful, the fungi?

The sentence needs to be clarified.

Figure 5:

Why, the measuring points are reduced?

Discussion: Lines 256/257

Without exact specification, it is not possible which species of the T. mentagrophytes complex is correct.

Reviewer 3 Report

Please see detail comments.

The MS entitle"Extended use of topical efinaconazole remains safe and can provide continuing benefits for dermatophyte toenail onychomycosis" provide useful information regarding phase IV clinical trial of Efinaconasol.Below some of comments/questions would be addressed by authors.

Introduction:in Line 41, explain the causative agents of onycomycosis and clarify what species are more resistant to exiting antifungal drugs commonly used for treatment of onycomycosis with therapeutic failure and poor outcome.This clarify the importaance of development of   new antifungal drugs efficiently.

Material:

-What was the control group in this trial?

-Line 92, write full name of CG agar with relevant reference.

-  what is double-blind vehicle control in line 74? 

Result:

-Remove all background lines in figures.

-For patients who experienced adverse effects and had to discontinue efinaconazole, which alternative medications were used for their treatment?

-Please clarify line 184.

-Table1. What  method did you use to identify Dermatophayte spp  in species level?Please rewrite line 124-126.

How do you interpret the adverse events in patients mentioned in Table 2? might be related to some predisposing factors I those patients that trigger the side effects of efinaconazole?

Is there any Mycological data regarding MIC of efinaconazole against different Dermatophayte spp?it was fungicide or fungistatic?

How do you explain the increased cure rate from M12 to M24 in patients?

Round 2

Reviewer 2 Report

Congrats, my questions were answered successfully

no further comments